# Caregivers’ and Family Members’ Knowledge Attitudes and Practices (KAP) towards Epilepsy in Rural Limpopo and Mpumalanga, South Africa

**DOI:** 10.3390/ijerph20065222

**Published:** 2023-03-22

**Authors:** Ofhani Prudance Musekwa, Lufuno Makhado, Angelina Maphula

**Affiliations:** 1Department of Psychology, Faculty of Health Sciences, University of Venda, Thohoyandou 0950, South Africa; 2Faculty Deans Office, Faculty of Health Sciences, University of Venda, Thohoyandou 0950, South Africa

**Keywords:** attitude, caregiver, epilepsy, family member, knowledge, practices

## Abstract

Epilepsy, a common neurological disease, has a significant impact on people living with epilepsy (PLWE), their caregivers, and their family members. Research has consistently shown that the quality of life of PLWE is low. To expand on this knowledge, a non-experimental quantitative survey study was conducted to explore the knowledge, attitudes, and practices (KAP) of caregivers and family members towards epilepsy and epilepsy-related seizures. The study sample consisted of 519 participants from two South African provinces (Limpopo and Mpumalanga), mostly aged 26–35 years. The study revealed that most respondents in Limpopo had no formal education, whereas in Mpumalanga, most had a secondary education. Most respondents (32.4%) reported always using a spoon to prevent tongue biting during seizures. However, 62.4% of respondents reported feeling unprepared to handle an epileptic seizure. Additionally, the majority (54.7%) showed a moderate level of knowledge about epilepsy. Many respondents had a negative attitude towards epilepsy, and there was uncertainty about proper practices during a seizure. In summary, the research highlights unsatisfactory knowledge and practices towards epilepsy and emphasizes the need for increased education and awareness among caregivers and family members. Significant educational investment is needed from medical services to improve epilepsy care, knowledge, and attitudes.

## 1. Introduction

Globally, about 50 million people have epilepsy, making it one of the world’s most common diseases. In sub-Saharan Africa, 9 in 1000 people live with epilepsy [1]. However, [2] argues that there may be a significant difference in the prevalence of epilepsy based on factors such as the type of epilepsy accounted for. Thus, these statistics may not reflect the prevalence of epilepsy, and statistics may be higher. 

With these statistics in mind, it is crucial to understand that when an individual is living with epilepsy, it does not only affect them but also their family members and caregivers. Subsequently, family members and caregivers’ lives are altered and, to some degree, burdened by living with or caring for a person with epilepsy [3,4]. This may influence practices toward PLWE, especially during a seizure, because it was found that about 80% of Africans consult traditional healers and engage in traditional rituals in an attempt to stop the seizure or cure epilepsy [5]. Some of these rituals include herb steaming and using *gonono* (Black coloured insect) and plant mixtures [5]. This supports the idea that there is a need to provide informational support to family members and caregivers [6]. This information is expected to increase care ability, reduce burden, and increase knowledge as well as awareness toward epilepsy patient needs, care options, and risks.

Amongst other studies, studies [7,8,9,10,11] show that most literature on caregivers of people living with epilepsy tends to focus more on the epilepsy burden, primarily on children/teenagers living with epilepsy. On the other hand, [12] conducted a first-aid intervention study in the UK, and on many occasions, we found that initiatives such as these are specific to developed countries. However, because family members act as life-saving agents for PLWE, their knowledge, awareness, and practice towards PLWE are pivotal to the quality of life and overall life experience of the PLWE (regardless of social class) [13]. 

This makes it essential for them to have good knowledge of epilepsy and epilepsy first aid. This study aimed to explore family members’ and caregivers’ knowledge, practices, and awareness of epilepsy in a rural setting in Limpopo. The researchers found it important to investigate the KPA of caregivers of PLWE of all ages.

## 2. Materials and Methods

This study was a non-experimental quantitative survey conducted in rural South Africa, Limpopo, and Mpumalanga province. The two provinces and villages were purposefully selected (according to the cultural representativeness of South Africa). Mtititi, Malavuwe, Nweli, and Bochum were villages selected in Limpopo; Clare, Acornhoek, and Jerusalem were selected in Mpumalanga. Home-based carers (HBCs) purposefully sampled study respondents based on families they already knew had patients LWE. Households consulted had one known patient LWE. Total population sampling of known families (not on official records) was applied. From all the families, 519 respondents agreed to participate in the study, and 15 refused to participate. Every potential respondent had an opportunity to consent to participate in the study. Only individuals above 18 who were caring for (as the primary caregiver or individuals who sometimes care for PLWE) or are a family member to a person LWE were considered eligible for participation. All family members within the household were approached for participation. Before data collection, the royal authorities of the mentioned villages and the Department of Health gained permission to conduct the study. On approval from the royal authorities was granted, the researchers trained the HBCs on administrating the questionnaire for respondents who could not read/write. Literate respondents were given a printed questionnaire with a sanitized pencil to take the questionnaire on their own. 

For ease of understanding, the data collection tool was translated from English to Tshivenda, Sehananwa, Xitsonga, and Seswati. The questionnaire comprised three sections; Section A: demographic profile; Section B: knowledge and attitude toward epilepsy; and Section C: practices towards epilepsy. Regarding the total knowledge score, the questionnaire had 26 questions to measure knowledge. Each correct answer scored a 1, and the incorrect answer was 0. The cores 1–8 were considered low; 9–17 were considered moderate; 18 and above were considered high knowledge. With regard to attitudes, the researchers measured attitudes based on their positive or negative responses to the questions. There were six questions measuring attitudes from respondents. Zero was assigned to each negative response, and 1 to the positive response. 

The collected data was analysed using Statistical Package of Social Sciences (SPSS) software version 26.0. The researchers conducted a Chi-square cross-tabulation per province on demographic data and practices toward epileptic seizures and PLWE, descriptive and cross-tabulation on knowledge, and cross-tabulation on awareness. The study was conducted according to the guidelines of the Declaration of Helsinki and approved by the Institutional Review Board (or Ethics Committee) of the University of Venda (SHS/20/PSYCH/12/2710, approved 30 October 2020).

## 3. Results

### 3.1. Study Demographics

This current study included N = 519 respondent family members. Most respondents were from Mpumalanga (*n* = 280), and Limpopo had a total of *n* = 239 respondents. As illustrated in Table 1, most findings show that in both provinces the results were highly significant. Furthermore, most respondents were individuals aged 18–25 (Limpopo 33.9%, Mpumalanga 36.1%) and female (Limpopo 69.9%, Mpumalanga 78.2%). In addition, most respondents were Christian faith believers. Limpopo had 65.7% and the minority was Islam faith believers (0.8%); Mpumalanga had 91.4% of Christian faith believers as majority respondents and 0.4% were the Muslim minority. Regarding employment status, from both provinces, most respondents were unemployed (87.4% in Limpopo and 76.41 in Mpumalanga) and depended on social grants reliant (68.0% in Limpopo and 61.8% in Mpumalanga). More Northern Sotho/Pedi respondents were from Limpopo (59%), 25.1% were Tsonga speaking, and 11% were Venda. Whereas Mpumalanga had more Tsonga respondents (77.9%), 18.2% were Swati, and 2.9% were Northern Sotho/Pedi respondents. Regarding the relationship to the PLWE, in the study, it was found that in Limpopo, the majority of respondents were parents of the patient (31.3%), followed by siblings (19.25%) and nieces/nephews (14.2%). In Mpumalanga, the highest number of respondents were the patients’ siblings (27.5%), followed by parents (25.7%) and children (19.6%). When it comes to the distance to the nearest primary health facility, 41.4% of those in Limpopo travel 10–19 km to go to local clinics, while 77.5% of those in Mpumalanga travel 0–9 km. In Limpopo, 35.6% of the respondents travel 0–9 km to a hospital, while 15.8% travel 10–19 km.

### 3.2. Practices toward Epilepsy

Table 2 shows that during an epileptic attack, most respondents sometimes (33.1%) and always (32.4%) put a spoon to prevent tongue biting. Many respondents (47.2%) from both provinces said they never prayed for the seizures to stop. However, 39.7% in Limpopo and 39.1% in Mpumalanga sometimes pray to stop seizures. According to the survey, the majority of respondents (82.9%) do not cover patients with a blanket during an epileptic seizure, while 11.9% use a blanket occasionally. Caregivers sometimes restrain the patient during a seizure by holding their legs (39.7%), while 32.2% always restrain the patient. From both provinces, 6.4% constitutes of respondents who said they always spray holy water or oil on a person having an epileptic seizure. Considering treatment, most respondents (75.5%) showed that no family traditional doctor treats PLWE. The study found that a majority of respondents, 81.5%, never perform traditional rituals during a seizure, while only 14.1% do so occasionally. Interestingly, a significant percentage of respondents, 53.4% from both provinces, stated that there is no western doctor who treats people living with epilepsy, while 24.3% said that a western doctor always treats the patient. Moreover, 55.9% of respondents reported that they sometimes call for medical assistance when an epileptic seizure occurs.

### 3.3. Caregiver/Family Member Knowledge

As shown in Table 3, the current study found that most respondents from both provinces had moderate knowledge of epilepsy (54.7%), with a mean of 16.57 and 3.34 standard deviation. It was found that 43.7% of respondents have a good understanding of epilepsy. However, it should be emphasized that the knowledge score is significantly influenced by the responses from Limpopo province, where 63.2% of respondents have a moderate level of knowledge. Conversely, in Mpumalanga, the majority of respondents have a high level of knowledge (51.4%). In Limpopo, 32.2% of respondents agreed that they and their family have received information about epilepsy, while 16.7% disagreed (see Table 4). In contrast, in Mpumalanga, 33.6% disagreed that they have received information about epilepsy, while 30.4% agreed. Furthermore, 40.6% of respondents in Limpopo reported having access to information about epilepsy, while 37.4% of those in Mpumalanga province reported the same.

### 3.4. Caregiver/Family Member Attitude

After cross-tabulating questions on attitude (see Table 5), we found that respondents’ negative and positive attitudes were revealed among the questions raised. According to the findings, most respondents showed to have negative attitudes toward epilepsy from both provinces. The majority of the respondents from both Mpumalanga (86.1%) and Limpopo (74.9%) believed that a person LWE should not marry, and 74.6% (Mpumalanga) and 67.4% (Limpopo) stated that they are not allowed to do chores. Positive attitudes were shown when respondents were asked whether people living with epilepsy (PLWE) are a danger to themselves, with 57.7% of respondents in Limpopo and 76.4% in Mpumalanga agreeing that PLWE are not a danger to others. On the other hand, 57.7% of respondents in Limpopo and 76.4% in Mpumalanga disagreed with the notion that PLWE are a danger to others, demonstrating a positive attitude towards epilepsy. Additionally, when asked if PLWE should be segregated from the rest of the family members, 79.1% of respondents in Limpopo and 94.3% in Mpumalanga expressed a positive attitude towards epilepsy.

## 4. Discussion

Findings from this study show that, in most instances, the primary caregiver is the patient’s mother, who, in most instances, is unemployed [13,14]. This may be because caregivers felt they needed to stay at home with PLWE to care for them better, which may have resulted in them quitting their jobs. Consequently, unemployment increases caregiver and family frustration and burden [15,16]. However, in Mpumalanga, the findings showed an equal number of mothers to a brother that is a primary caregiver to PLWE. This finding may support the claim that caring for a person’s LWE may result in lowered quality of life for the caregiver and siblings [17]. How much more would this impact be if the sibling is also the primary caregiver?

With the lowered quality of life, the financial burden also contributes. As the findings of this study showed, most caregivers relied on social grants for support. This may result in low-quality treatment and the ability to seek the best treatment for the patient, especially because the study showed that most homes were further from hospitals compared to primary health care. However, regardless of the distance, caregivers endure and take their patients to western facilities, not traditional facilities for treatment. As the results show, that is the preference of the respondents. Many respondents said they never take the PLWE for traditional treatment; this does not coincide with some findings [18]. Furthermore, the current study’s findings argue that one’s religious beliefs do not influence the caregiver’s choice of treatment, as many stated that they do not put oil to stop the seizure, and the majority call for medical assistance.

However, the findings show respondents may not be well-equipped with epileptic seizure first aid. There is uncertainty regarding what to do during an epileptic seizure. Many reported that they sometimes or always feel unprepared to manage epileptic seizures; they pray for the seizure to stop, restrain the person’s legs, and call on the person to assess and regain consciousness. These practices may result in fear and anxiety, and they may exhibit such behaviour due to the build-up of fear and anxiety following seizure frequency and training. Moreover, a significant majority of respondents reported frequently calling for medical assistance during a seizure, which may suggest the severity of seizures experienced by patients in these provinces. This finding could also contribute to caregivers feeling ill-equipped to handle seizures, particularly in terms of providing first aid. As stated in studies conducted [19,20,21], amongst other psychological impacts, anxiety is common in caregivers of PLWE.

Regarding caregiver and family members’ knowledge, it is unfortunate that the level of knowledge influences practices toward epileptic seizures and our knowledge sharing about epilepsy. In this study, the total knowledge score is worrisome. Many respondents were found to have a moderate level of knowledge, especially when many claim to have had someone speak to them about epilepsy and have access to information about epilepsy. This leaves the researchers curious about the validity of the content and source of information about epilepsy. Nonetheless, the low levels of knowledge observed in the study may be linked to the respondents’ level of education, as this can influence their ability to comprehend and effectively apply information. 

In addition, this study shows that there may be a connection between caregiver knowledge and attitude. What we know will eventually impact how we think about epilepsy and influence those around us concerning epilepsy. As illustrated in the findings, many respondents showed a negative attitude toward epilepsy [22]. For this reason, active epilepsy awareness and information sharing should be conducted by healthcare providers, particularly those who interact with patients and their families for the first time. It is essential to extend this to the entire family, not just the primary caregivers. This will help improve attitudes towards epilepsy, enhance epilepsy care, and ultimately improve the quality of life for people living with epilepsy, their caregivers, and their families.

## 5. Conclusions, Limitations and Recommendations

We believe that while the caregiver burden faced by family members and caregivers of PLWE is an important issue, their practices, level of awareness, and knowledge also require significant attention. Despite being burdened, particularly financially, caregivers’ lack of information on proper practices towards epileptic patients, and their negative attitudes and knowledge can jeopardize the lives of PLWE and negatively impact their quality of life. Therefore, this study’s findings suggest the need for healthcare workers and researchers to receive financial support and culturally congruent education on PLWE care and first aid to address the issue. One of the limitations of this study is that it solely relied on quantitative methods for data collection. Furthermore, the survey tool did not evaluate particular aspects such as seizure triggers, warning signs, and seizure recovery as observed by caregivers. Another limitation is that the study was carried out in only two provinces of South Africa, and although the residents are representative of the country’s populations and cultures, the sampled respondents who agreed to participate were not diverse enough to fully represent the range of cultures that exist within the provinces. In future studies, researchers could focus on exploring how the knowledge and practices of caregivers impact the quality of life of PLWE. Another area of interest could be the role of formal education in enhancing KAP and the causes of negative attitude towards epilepsy. While the current study found that respondents had access to information on epilepsy, it would be beneficial to investigate the type and source of information as well as the frequency of knowledge sharing among caregivers and family members. Such research could provide valuable insights into how to improve education and support for caregivers and families of PLWE, ultimately leading to better outcomes for those living with epilepsy.

## Figures and Tables

**Table 1 ijerph-20-05222-t001:** Respondents’ demographic data.

		Province				X^2^
		Limpopo		Mpumalanga		(*p*-Value)
Age	18–25 years	81	33.9%	101	36.1%	4.047 (*p* = 0.543)
	26–35 years	56	23.4%	66	23.6%	
	36–45 years	36	15.1%	40	14.3%	
	46–55 years	23	9.6%	30	10.7%	
	56 and above	43	18%	43	15.4%	
Gender	Female	167	69.9%	219	78.2%	5.559 (*p* = 0.062)
	Male	71	29.7%	61	21.8%	
	Other	1	0.4%	0	0.0%	
Ethnic group	Northern Sotho/Pedi	141	59.0%	8	2.9%	283.854 (*p* < 0.001)
	Swati	3	1.3%	51	18.2%	
	Ndebele	1	0.4%	0	0.0%	
	Venda	28	11.7%	1	0.4%	
	Tsonga	60	25.1%	218	77.9%	
	Afrikaans	1	0.4%	0	0.0%	
	Zulu	0	0.0%	1	0.4%	
	SeSotho	5	2.1%	0	0.0%	
	Other	0	0.0%	1	0.4%	
Highest educational qualification	No formal education	70	29.3%	43	15.4%	27.042 (*p* < 0.001)
	Primary education	61	25.5%	69	24.6%	
	Secondary education with no Grade 12	69	28.9%	74	26.4%	
	Secondary education with Grade 12	29	12.1%	74	26.4%	
	Tertiary education	10	4.2%	20	7.1%	
Religion	Christianity	157	65.7%	256	91.4%	52.690 (*p* < 0.001)
	Traditional	80	33.5%	23	8.2%	
	Islam	2	0.8%	1	0.4%	
Employment or work status	Not employed	209	87.4%	214	76.4%	12.353 (*p* > 0.001)
	Self-employed	18	7.5%	37	13.2%	
	Employed	12	5.0%	29	10.4%	
Current source/s of income	Formal employment	12	5.0%	30	10.7%	14.229 (*p* > 0.001)
	Social grant	165	69.0%	173	61.8%	
	Self-employment	18	7.5%	38	13.6%	
	Other	44	18.4%	39	13.9%	
Relationship with PLWE	Parent	75	31.3%	72	25.7%	68.008 (*p* < 0.001)
	Child	29	12.1%	55	19.6%	
	Sibling	46	19.2%	77	27.5%	
	Niece/nephew	34	14.2%	11	4.0%	
	Aunt/uncle	17	7.1%	15	5.4%	
	In law	7	3.0%	6	2.2%	
	Grandchild	8	3.3%	15	5.4%	
	Grandparent	5	2.1%	1	0.4%	
	Cousin	12	5.0%	10	3.6%	
	Spouse	6	2.5%	4	1.5%	
	Caregiver	0	0.0%	12	4.3%	
	Romantic partner	0	0.0%	2	0.7%	
Distance of your household to the nearest PHC Facility/Clinic.	0–9 km	85	35.6%	217	77.5%	104.056 (*p* < 0.001)
	10–19 km	99	41.4%	44	15.8%	
	20–29 km	28	11.7%	16	5.7%	
	30–39 km	25	10.5%	2	0.7%	
	40–49 km	0	0.0%	1	0.4%	
	50 km and above	2	0.8%	0	0.0%	
Distance of your household to the nearest Hospital.	0–9 km	10	4.2%	6	2.2%	90.997 (*p* < 0.001)
	10–19 km	76	31.8%	30	10.8%	
	20–29 km	81	33.9%	65	23.3%	
	30–39 km	41	17.2%	65	23.3%	
	40–49 km	9	3.8%	89	31.9%	
	50 km and above	22	9.2%	24	8.6%	

**Table 2 ijerph-20-05222-t002:** Practices toward epileptic seizures.

		Province				Total		X^2^
		Limpopo		Mpumalanga				(*p*-Value)
During epileptic seizures, I put a steel spoon in the mouth to prevent tongue bite.	Never	118	49.4%	61	21.8%	179	34.5%	48.94 (*p* < 0.001)
	Sometimes	72	30.1%	100	35.7%	172	33.1%	
	Always	49	20.5%	119	42.5%	168	32.4%	
During epileptic seizures, I pray for the seizures to stop.	Never	117	49.0%	128	45.7%	245	47.2%	2.17 (*p* = 0.33)
	Sometimes	95	39.7%	108	38.6%	203	39.1%	
	Always	27	11.3%	44	15.7%	71	13.7%	
I spray holy water or oil on them to stop the seizure	Never	167	69.9%	248	88.6%	415	80.0%	28.74 (*p* < 0.001)
	Sometimes	51	21.3%	20	7.1%	71	13.7%	
	Always	21	8.8%	12	4.3%	33	6.4%	
There is a traditional family doctor who treats the PLWE in our family	Never	163	68.2%	229	81.8%	392	75.5%	12.96 (*p* < 0.001)
	Sometimes	61	25.5%	42	15.0%	103	19.8%	
	Always	15	6.3%	9	3.2%	24	4.6%	
There is a medical doctor who treats the PLWE in our family	Never	98	41.0%	179	63.9%	277	53.4%	31.07 (*p* < 0.001)
	Sometimes	60	25.1%	56	20.0%	116	22.4%	
	Always	81	33.9%	45	16.1%	126	24.3%	
I cover the person with a blanket during an epileptic attack	Never	167	69.9%	263	93.9%	430	82.9%	52.79 (*p* < 0.001)
	Sometimes	49	20.5%	13	4.6%	62	11.9%	
	Always	23	9.6%	4	1.4%	27	5.2%	
During epileptic seizures, a traditional ritual is performed	Never	173	72.4%	250	89.3%	423	81.5%	24.59 (*p* < 0.001)
	Sometimes	51	21.3%	22	7.9%	73	14.1%	
	Always	15	6.3%	8	2.9%	23	4.4%	
During epileptic seizures, I pour cold water over the person.	Never	130	54.4%	207	73.9%	337	64.9%	21.86 (*p* < 0.001)
	Sometimes	80	33.5%	56	20.0%	136	26.2%	
	Always	29	12.1%	17	6.1%	46	8.9%	
During epileptic seizures, I restrain the person by holding the legs	Never	70	29.3%	76	27.1%	146	28.1%	0.46 (*p* = 0.813)
	Sometimes	95	39.7%	111	39.6%	206	39.7%	
	Always	74	31.0%	93	33.2%	167	32.2%	
During epileptic seizures, I call for medical help	Never	53	22.2%	64	22.9%	117	22.5%	2.35 (*p* = 0.31)
	Sometimes	141	59.0%	149	53.2%	290	55.9%	
	Always	45	18.8%	67	23.9%	112	21.6%	
During epileptic seizures, I do not put anything in the mouth of the person.	Never	97	40.6%	129	46.1%	226	43.5%	9.59 (*p* < 0.001)
	Sometimes	93	38.9%	121	43.2%	214	41.2%	
	Always	49	20.5%	30	10.7%	79	15.2%	
During epileptic seizures, I call the person by name assess consciousness	Never	70	29.3%	77	27.5%	147	28.3%	0.26 (*p* = 0.88)
	Sometimes	92	38.5%	113	40.4%	205	39.5%	
	Always	77	32.2%	90	32.1%	167	32.2%	
I feel unprepared to manage epileptic seizures.	Never	83	34.7%	112	40.0%	195	37.6%	3.53 (*p* = 0.17)
	Sometimes	102	42.7%	97	34.6%	199	38.3%	
	Always	54	22.6%	71	25.4%	125	24.1%	
In our family, a PLWE is given traditional medication to control seizures.	Never	143	59.8%	177	63.2%	320	61.7%	2.37 (*p* = 0.31)
	Sometimes	69	28.9%	65	23.2%	134	25.8%	
	Always	27	11.3%	38	13.6%	65	12.5%	

**Table 3 ijerph-20-05222-t003:** Caregiver/family member Total Knowledge Score.

		N		Min-Max		Mean (SD)		
Total Knowledge Score		519		6.00–24.00		16.57 (3.34)		
		Province				Total		X^2^ (*p*-value)
		Limpopo		Mpumalanga				
Total Knowledge Score	Low Knowledge Level	5	2.1%	3	1.1%	8	1.5%	14.89 (*p* = 0.001)
	Moderate Knowledge Level	151	63.2%	133	47.5%	284	54.7%	
	High Knowledge Level	83	34.7%	144	51.4%	227	43.7%	

**Table 4 ijerph-20-05222-t004:** Caregiver/family member knowledge of epilepsy.

		Province				X^2^ (*p*-Value)
		Limpopo		Mpumalanga		
Someone has spoken to me and my family about epilepsy.	Strongly agree	77	32.2%	81	28.9%	16.73 (*p* = 0.001)
	Agree	69	28.9%	85	30.4%	
	Disagree	53	22.2%	94	33.6%	
	Strongly disagree	40	16.7%	20	7.1%	
My family and I have access to information on epilepsy.	Strongly agree	74	31.0%	85	30.4%	2.12 (*p* < 0.001)
	Agree	97	40.6%	104	37.1%	
	Disagree	45	18.8%	67	23.9%	
	Strongly disagree	23	9.6%	24	8.6%	

**Table 5 ijerph-20-05222-t005:** Caregiver/family member attitudes toward epilepsy.

		Province				X^2^ (*p*-Value)
		Limpopo		Mpumalanga		
Epilepsy is taboo and should not be publicly spoken about.	Negative attitude	160	66.9%	210	75.0%	4.08 (*p* < 0.001)
	Positive attitude	79	33.1%	70	25.0%	
A person living with epilepsy should not marry.	Negative attitude	179	74.9%	241	86.1%	10.43 (*p* = 0.001)
	Positive attitude	60	25.1%	39	13.9%	
People living with epilepsy are a danger to others.	Negative attitude	101	42.3%	66	23.6%	20.63 (*p* > 0.001)
	Positive attitude	138	57.7%	214	76.4%	
PLWE are not allowed to perform work activities like other family members.	Negative attitude	161	67.4%	209	74.6%	3.33 (*p* < 0.001)
	Positive attitude	78	32.6%	71	25.4%	
A person living with epilepsy is not allowed to sit near fires or cook in our family.	Negative attitude	148	61.9%	183	65.4%	0.58 (*p* < 0.001)
	Positive attitude	91	38.1%	97	34.6%	
A person living with epilepsy is separated from the rest of the family.	Negative attitude	50	20.9%	16	5.7%	26.86 (*p* > 0.001)
	Positive attitude	189	79.1%	264	94.3%	

## Data Availability

Not applicable.

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
