# Peer review of "Caregivers’ and Family Members’ Knowledge Attitudes and Practices (KAP) towards Epilepsy in Rural Limpopo and Mpumalanga, South Africa"

_ijerph, 2023, doi:10.3390/ijerph20065222_

Round 1
Reviewer 1 Report
The paper deals with a relevant issue to public health. However, there is a set of questions which I feel need clarifying and must be considered before publication, that support my recommendation that the papers needs major revision.
1) The title of the paper is not completely clear to international readers and readers from different scientific fields. Being an interdisciplinary journal, the acronym KAP is not self-evident for all disciplinary areas. Besides that, if the authors consider it important to identify the research location in the title, they should also include the country.
2) Abstract: The country should be identified before provinces are mentioned. There is no point in having the mean value of the knowledge level without including the range, I suggest removing it.
3) Materials and Methods: The country should be identified before provinces are mentioned. It is necessary to explain the criteria for the selection of the provinces and, within these, of the villages. It is important to mention how many people were eligible to participate in the study and how many did not give their consent. It is also important to mention how many persons with epilepsy the respondents correspond to. Is there only one caregiver/family member per person with epilepsy or more? The authors mention that the researchers trained the HBCs on how to administrate the 65 questionnaire for respondents who cannot read/ write. How about the respondents who can read and write? How was the questionnaire administrated?
4) Results: In 3.1. there should be an explicit reference to “ethnic group” in lines 90-92. There is an error in the presentation of many percentages, which are preceded by “n=”. If it is a percentage, it cannot be n (this is a constant error throughout the presentation of the results).
3.2., 3.3. and 3.4.: In the results presentation, the only comparison that is made is between the two provinces. The authors do not analyze the practices, knowledge and attitudes taking into account other variables that will certainly influence the KAP, such as the level of education. Nor do they establish correlations between the level of knowledge and attitudes and practices. The authors vaguely mention this relation as a hypothesis at the end of the Discussion, but do not provide empirical evidence of this relation, which seems a pity, since they have the data to do so. Overall, the article would benefit from further exploration of the data.
3.2.: In the text, the logic of the presentation of the results is unclear. Some practices are highlighted and not others that have the same quantitative expression. What is the logic behind highlighting this data?
3.3. The knowledge score needs to be explained; how was the score built? Based on which indicators? How were the three levels of knowledge defined?
3.4.: It is necessary to explain how the authors come to classify attitudes as positive or negative. What is the scale for responses to the assertions in Table 4?
5) The paper requires an extensive editing of English language and style. There are expressions that are too colloquial for a scientific paper, there are words that are not suited to the meaning the authors intend, there are prepositions used incorrectly, etc. Some examples:
(line 82) “most findings show that in both provinces showed…” – repetition
(line 83) “Furthermost of the respondents were individuals…” – “Most”?
(line 87) “had 91.4% of Christian faith believers as majority respondents” – “majority respondents” seems redundant
(line 103) “6.4% constitutes of respondents said they always” - the sentence is grammatically incorrect
(line 143) “that most homes were further from hospitals” – “far”?
(line 170) “We are of the impression” – too colloquial
Author Response
Dear Reviewer.
Thank you so much for your input and suggestions; they were very helpful in shaping the manuscript and making it even better than the original submission. Your ideas and corrections are very appreciated.
See attached document for your comments and recommendations.
Regards,
Authors

Reviewer 2 Report
Ok, a lot of work has gone into this paper. As a westerner I am fascinated by some of the cultural content.
It is rather 'wordy' and I feel needs considerable 'trimming'. For example the title doesn't need 'I should probably know how to do this' in the title. What does is KAP in the title mean?
I don't need to know about prevalance in America or Asia.
What is accepted as 'standard' first aid in SA and say Europe?
Is this study comparing first aid, two different districts, or attitudes to epilepsy. There is almost 2 papers here, first aid and attitude. Comparing two different social areas with different educational attainments. No mention of the 'recovery position' or timing. Education is lacking, perhaps no need to compare the two districts as the 'issues' appear in both, it's not surprise they appear in the poorer less educated area.
It would be interesting in the introduction to know what 'traditional' medicine and ritual entails. You would expect 60-70% PWE to be controlled on western medication. Are all the respondents caring for PWE with poor control?
There is nothing re limitations in the study, and more importantly nothing about 'what can be done to improve the situation'. Education is lacking, who is going to educate, schools, nurses, docs. How do we get past religion and tradition?
Author Response

(The authors gave the same response as above.)

Round 2
Reviewer 1 Report
The changes that were introduced in the paper improved considerably its quality. My recommendation is that the paper should be accepted for publication in its present form.
Author Response
Dear Reviewer,
We appreciate your interest and contributions. Your recommendations have been of great value to ensure the quality of the manuscript is good and acceptable.
Thank you greatly.
Authors.
Reviewer 2 Report
I still think it's too long, i dont need to know you used sps in the abstract. The limitations and 'where we go from here', future study, is too brief.
Make the paper a 1/3 less wordy.
There is a 'lack of knowledge' and 'negative attitude' - what do you suggest is done. Maybe nothing can be done because of reliance on traditional medicine. There is nothing in the abstract to reflect what needs to be done, education and investment?
You have shown a difference in attitude between two different areas, what does this add to our knowledge of epilepsy care. I could do the same for say inner city working class Liverpool and a more leafy middle class suburb and get the same result. It's what we do about it that is more important. I though wouldn't be 'up against' traditional medicine. I assume this is the same for other illnesses other than epilepsy.
The english is fine, just too much. See below for a rough edit on your abstract (took me 5 mins).
Epilepsy is a common neurological condition that has a negative impact on People Living With Epilepsy (PLWE),their caregivers and family members. This study sought to discover caregivers' and family members' knowledge, awareness, and practices (KAP) toward patients with epilepsy and epilepsy-related seizures. This non-experimental quantitative survey studied 519 respondents, in two provinces, Limpopo, where most had no formal education, and Mpumalanga, where most had secondary education. (Most respondents (32.4%) always put a spoon to prevent tongue biting. In addition, 62.4% of respondents reported that they sometimes and always feel unprepared to handle an epileptic seizure when it happens. Regarding total knowledge score, 54.7% (majority) showed a moderate level of knowledge (on 16.57 mean and 3.34 standard deviation). Is this the best summary?? Many (figure) respondents showed to have a negative attitude toward epilepsy. The level of knowledge regarding epilepsy was unsatisfactory, and there is uncertainty in proper practices during an epileptic seizure. Significant educational investment is needed from medical services to improve care.
Author Response
Dear Reviewer,
We appreciate your candid feedback, and we have gained valuable insights from your comments and recommendations that we will incorporate into our future manuscripts.
We aim to ensure that the received manuscript meets your expectations. Additionally, please take note of the rebuttal that is attached.
Regards,
Authors.
